# The role and mechanism of tetramethylpyrazine for atherosclerosis in animal models: A systematic review and meta-analysis

**SiJin Li**[1,2], **Ping Liu**[1,2]*, **XiaoTeng Feng**[1,2], **YiRu Wang**[1,2], **Min Du**[1,2], **JiaRou Wang**[1,2]

**1** Longhua Hospital, Shanghai University of Traditional Chinese Medicine, Shanghai, China, **2** Shanghai University of Traditional Chinese Medicine, Shanghai, China

* liuping0207@yeah.net

**Data Availability Statement:** All relevant data are within the paper and its Supporting Information files.

## Abstract

### Background

Atherosclerosis(AS) is widely recognized as a risk factor for incident cardiovascular and cerebrovascular diseases. Tetramethylpyrazine (TMP) is the active ingredient of Ligusticum wallichii that possesses a variety of biological activities against atherosclerosis.

### Objective

This systematic review and meta-analysis sought to study the impact of and mechanism of tetramethylpyrazine for atherosclerosis in animal models.

### Methods

A systematic search was conducted of PubMed, Embase, Cochrane Library, Web of Science database, Chinese Biomedical (CBM) database, China National Knowledge Infrastructure (CNKI), WanFang data, and Vip Journal Integration Platform, covering the period from the respective start date of each database to December 2021. We used SYRCLE's 10-item checklist and Rev-Man 5.3 software to analyze the data and the risk of bias.

### Results

Twelve studies, including 258 animals, met the inclusion criteria. Compared with the control group, TMP significantly reduced aortic atherosclerotic lesion area, and induced significant decreases in levels of TC (SMD = -2.67, 95% CI -3.68 to -1.67, $P < 0.00001$), TG (SMD = -2.43, 95% CI -3.39 to -1.47, $P < 0.00001$), and LDL-C (SMD = -2.87, 95% CI -4.16 to -1.58, $P < 0.00001$), as well as increasing HDL-C (SMD = 2.04, 95% CI 1.05 to 3.03, $P = 0.001$). TMP also significantly modulated plasma inflammatory responses and biological signals associated with atherosclerosis. In subgroup analysis, the groups of high-dose TMP ($\geq$50 mg/kg) showed better results than those of the control group. No difference between various durations of treatment groups or various assessing location groups.

**Funding:** This work was supported by grants from the National Natural Science Foundation of China (NSFC. 82074200, 81873117). The funders had no role in study design, data collection, and analysis, decision to publish, or preparation of the manuscript.

**Competing interests:** The authors have declared that no competing interests exist.

## Conclusion

TMP exerts anti-atherosclerosis functions in an animal model of AS mediated by anti-inflammatory action, antioxidant action, ameliorating lipid metabolism disorder, protection of endothelial function, antiplatelet activity, reducing the proliferation and migration of smooth muscle cells, inhibition of angiogenesis, antiplatelet aggregation. Due to the limitations of the quantity and quality of current studies, the above conclusions need to be verified by more high-quality studies.

## Trial registration number

PROSPERO registration no.CRD42021288874.

## Introduction

Atherosclerosis (AS) is a common chronic inflammatory progressive disease involving large and medium elastic and muscular arteries, which main pathological manifestations are high lipid accumulation, fibrous tissue hyperplasia, foam cells, and high inflammation [1]. AS is an essential pathological basis for inducing cardiovascular and cerebrovascular diseases [2]. The incidence of cardiovascular and cerebrovascular diseases related to it continues to rise and tends to be younger [3]. Related cardiovascular and cerebrovascular diseases, such as coronary heart disease, myocardial infarction, and stroke, are the leading causes of death and disability in the global population [4]. The treatment drugs mainly include statins, betas, niacin, and other lipid-regulating drugs, aspirin, clopidogrel, and other antiplatelet drugs, as well as thrombolytic anticoagulant drugs [5, 6]. Although these treatments have been effective in reducing LDL cholesterol levels to guidelines recommended, they have not been effective in reducing cardiovascular and cerebrovascular adverse events and carry risks of bleeding, liver and kidney damage, and rhabdomyolysis. Some studies have reduced cardiovascular and cerebrovascular adverse events caused by AS by immunomodulatory inhibition of IL-1β and confirmed that immune inflammation is involved in thrombosis of AS. However, these treatments will lead to an increased infection rate and thus increased mortality due to immunosuppression [7]. Therefore, it is of great clinical significance to actively explore the pathological mechanism of AS and effective treatment options.

Ligusticum chuanxiong Hort was first recorded in Shennong Ben Cao Jing. In traditional Chinese medicine (TCM), it is believed to have the function of tonifying blood, promoting blood circulation, and dissipating congestion [8]. TMP is the primary active substance of Ligusticum chuanxiong. Since the first isolation in 1957, accumulating interests have been focused on the effect of TMP on AS. Many studies have confirmed that TMP has many pharmacological effects, such as protecting the cardiovascular system, antiplatelet aggregation, and improving blood supply ability [9–11]. However, scattered evidence and uncertain mechanisms limit the clinical use of TMP in treating AS.

In this study, we evaluated the role and mechanism of TMP in atherosclerosis in animal models with a systematic review and meta-analysis to discover the clinical potential of TMP as an anti-atherosclerosis agent.

## Methods

This study was conducted following the updated Preferred Reporting Items for Systematic Reviews and Meta-Analyses (PRISMA) guidelines [12].

## Literature search

A systematic literature search was conducted using PubMed, Embase, Cochrane Library, Web of Science database, Chinese Biomedical (CBM) database, China National Knowledge Infrastructure (CNKI), WanFang data, and Vip Journal Integration Platform from their inception dates to December 2021. The key terms "ligustrazine", "chuanxiong-zine", "2,3,5,6-Tetramethylpyrazine", "tetramethylpyrazine-hydrochloride", "TMPZ", "atherosclerosis", " atherogenesis", "mouse", "mice", and "animals not humans" were used. Reference lists from the included articles were also searched to identify additional studies.

## Selection of studies

Two researchers independently conducted a systematic review according to the same eligibility criteria and included the study based on the agreement. When there was a disagreement or ambiguities, a third investigator joined and helped make the final decision.

## Eligibility criteria

**Types of studies.**    Controlled studies assessing the effects of ligustrazine on animal models with AS were searched. No language, publication date, or publication status restrictions were imposed. All case reports, clinical trials, reviews, and in vitro studies were excluded.

**Types of participants.**    Laboratory animal models of AS of any age, sex, or strain that were established in any manner were included. We excluded models with diabetes, hypertension, and other diseases.

**Types of intervention.**    The use of any type of TMP intervention compared with a placebo control was included. Placebo controls included equivalent amounts of non-functional substances (e.g., saline) or no treatment. Studies that combined multiple therapies were excluded.

**Types of outcome measure.**    The primary outcome measures were the histopathological analysis of the atherosclerotic lesion area. The second outcome measures were triglycerides (TG) or total cholesterol (TC) or low-density lipoprotein cholesterol (LDL-C), or high-density lipoprotein cholesterol (HDL-C). Other outcome measures were the related mechanisms of action in each study of TMP for AS.

## Data extraction

The following information of the included study was recorded: the surname of the first author, publication year, the details (species, gender, age, weight, number) of animals, AS model (method), information regarding treatment and control groups, intervention and dose, route, duration, anesthetic, measured outcomes, and information of TMP. In studies with multiple intervention arms, only data from the tetramethylpyrazine and negative control groups were considered in our analysis. We estimated the values from graphs using GetData Graph Digitizer 2.20 if the data were not described numerically in the study.

## Risk of bias in individual studies

Two authors used the Systematic Review Centre for Laboratory animal Experimentation (SYRCLE) 10-item quality checklist [13] to assess the risk of bias. The detailed criteria include: random sequence generation, baseline characteristics, allocation concealment, random housing, blinded interventions, random outcome assessment, blinding of outcome assessment, incomplete outcome data, selective reporting, and other bias. When there was a disagreement or ambiguities, a third investigator joined and helped make the final decision.

## Statistical analyses

Review Manager 5.3 software (provided by the Cochrane Collaboration) was used for meta-analyses and subgroup analyses. All outcome measures were considered continuous data, utilizing standardized mean difference (SMD) with a 95% confidence interval (CI) as the effect size. Heterogeneity between studies and subgroups was assessed using the Q-test and $I^2$ statistic. If $I^2 > 50\%$, the result was considered to have a high level of heterogeneity, then a random effect model was adopted. Instead, a fixed-effect model was used. Sensitivity analyses were conducted to test whether the preliminary results were robust. Graphpad Prism 8 was used to draw graphs. $P < 0.05$ was considered statistically significant.

Different dosages of TMP treated in the same study were defined as several independent experiments. We divided the animal numbers of the controlled group by the number of dosages to avoid an artificial increase in sample size.

## Results

### Study selection

The search identified 135 articles, of which 23 were duplicates and irrelevant articles. Titles and abstracts were screened, and 33 were excluded. After evaluating full-text articles, three studies were excluded due to lack of primary outcome; 13 studies were excluded because of vitro studies; 11 studies were excluded due to comparison with other traditional Chinese medical; 7 studies were excluded because of Combined with other medicine; 33 studies were excluded due to no AS model. Finally, this systematic review and meta-analysis identified twelve eligible studies. The specific search process is shown in Fig 1.

### Characteristics of the included studies

Detailed information regarding TMP in each study is displayed in Table 1. The characteristics of the twelve included studies are shown in Table 2. Seven studies [14, 16, 19, 21, 23–25] were published in Chinese, and five studies [15, 17, 18, 20, 22]were published in English between 1997 and 2021.

These studies involved 258 animals. Among them, five types of animals were used. Male ApoE-/- mice were used in four studies [16–19]; male Sprague Dawley (SD) rats were used in three studies [14, 20, 21]; male Wistar rats were used in two studies [23, 24]; male Ldlr-/- hamsters were used in one study [15]; male New Zealand white rabbits were used in one study [20]; both male and female rabbits were used in one study [25]. SD, Wistar rats, weighed 160–245g; rabbits weighed 0.85–3kg; mice weighed 18–27.3g, and the weight of Ldlr-/- hamsters was not reported. The sample size ranged from 4 to 12 animals in each group. Four studies [16, 17, 20, 24] established the AS model by feeding with high-fat and high-cholesterol diets for 12 weeks, eight weeks in two studies [15, 19], sixteen weeks in one study [18], ten weeks in one study [25], forty days in one study [23]. One study [14] established the AS model by feeding with high-fat diets for 12 weeks and intraperitoneal Vitamin D3 (2kg/ml). One study [21] achieved it by feeding with high-fat diets for eight weeks and intraperitoneal Vitamin D3 (70U/kg*4). One study [22] did so by feeding with atherogenic diets for six weeks and intraperitoneal Vitamin D3 (600,000IU/kg).

TMP doses varied among different studies. It mainly ranged from 20 to 200 mg/kg/d. In addition, 2 and 5mg/kg/d were used in two studies [16, 25]. Eight studies administered the TMP treatment via the intragastric route [15, 17, 18, 20, 22–25], and intraperitoneal injection in four studies [14, 16, 19, 21]. The duration of TMP treatment varied from 20 days to 12 weeks.

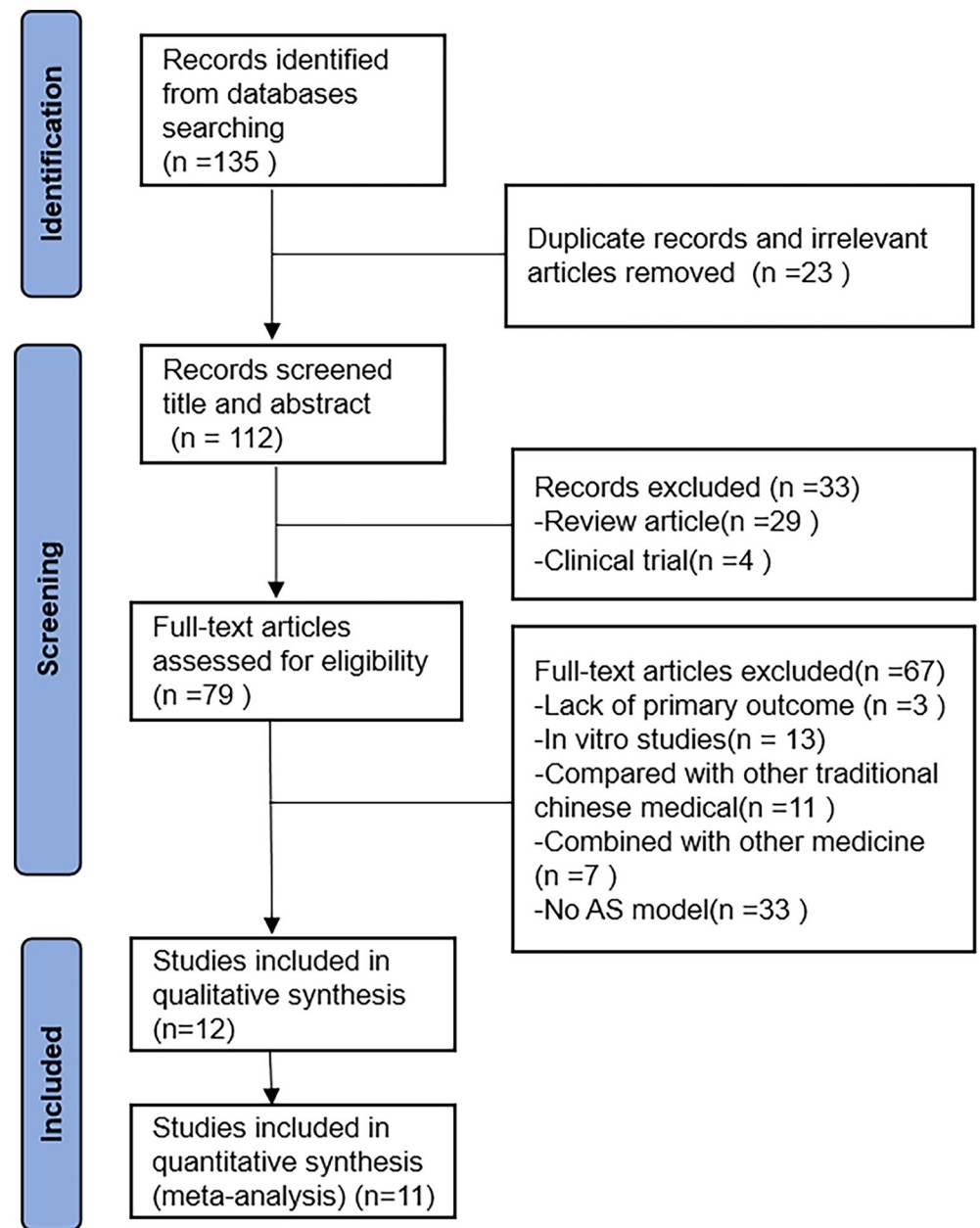

**Fig 1. Flow diagram of the study identification and selection process.**

Four studies [16, 23–25] did not mention using anesthesia. Five studies [15, 17, 19, 20, 22] used sodium pentobarbital. One study [14] used 10% chloral hydrate. One study [21] used urethane. One study [18] used CO2.

Regarding outcome measures, all studies reported atherosclerotic plaques. Eight studies reported quantitative analysis of lesion areas in aortas. Of these, four studies measured aortic lesion area at the aortic root level [15–18], and four studies measured lesion area within the whole aorta [19–22]. Four studies [14, 23–25] described the pathological morphology of the aortic lesion area. TC and TG were used in eight studies [14–17, 20, 22, 24, 25]. LDL-C and HDL-C were used in five studies [14, 16, 17, 20, 24]. HDL was used in two studies [15, 22].

**Table 1. Information on the TMP of each study.**

| Studies | Specifications | Source | Batch no. |
|---|---|---|---|
| Dong 2021 | Injection | China resources double crane | ? |
| Zhao 2020 | ? | ? | ? |
| Yuan 2019 | Tablet (100 mg) | Shanghai Yuanye Biotechnology Co., Ltd. | KM0513CA14 |
| Zhang 2017 | ? | ? | ? |
| Duan 2017 | Colorless acicular crystal (purity, 98.0%) | Sigma-Aldrich | ? |
| Ma 2015 | Tablet (80 mg) | Harbin Sanlian Pharmaceutical Co., Ltd. | 130302A1 |
| Wang 2013 | Tablet | Tong Ren Tang Company | ? |
| Dai 2013 | Colorless acicular crystal | Zhengzhou Zhuofeng Pharmaceutical Co., Ltd. | H2055479 |
| Jiang 2011 | Tablet | The Chinese National Institute for the Control of Pharmaceutical and Biological Products | ? |
| Lee 2009 | Tablet | Shanghai No.1 Biochemical Pharmaceutical Factory | 20070103 |
| Zhang 2005 | Tablet | NT | ? |
| Dong 1997 | Liposome10ml (containing TMP 2mg/kg) | Beijing Pharmaceutical Factory | ? |

TNF-α was used in two studies [14, 21]. phosphatidylinositol-3-kinase (PI3K) and phospho-threonine kinase (p-Akt) were used in two studies [15, 17]. Vascular endothelial growth factor (VEGF) and VEGFR-2 were used in two studies [14, 16]. Superoxide dismutase (SOD) was mentioned in two studies [19, 25]. Malondialdehyde (MDA) was mentioned in two studies [22, 25]. Furthermore, Interleukin-6 (IL-6), Interleukin-1β (IL-1β), cyclic adenosine monophosphate (cAMP), platelet activity, Akt, cluster of differentiation 31 (CD31), von Willebrand factor (vWF), Insulin level, progestin and adipoQ receptors 3 (PAQR3), SCAP/Sterol regulatory element binding protein 1c (SCAP/SREBP-1c), insulin receptor substrate 1 (IRS-1), mammalian target of rapamycin complex 1 (mTORC1), ATP-binding cassette transporters G1 (ABCG1), ATP-binding cassette transporters A1 (ABCA1), class A scavenger receptor (SR-A), cluster of differentiation 36 (CD36), glutathiones-transferase (GST), Nuclear factor E2 related factor 2 (Nrf-2), monocyte chemoattractant protein 1 (MCP-1), Intercellular Adhesion Molecule 1 (ICAM-1), oxidized low-density lipoprotein receptor 1 (LOX-1), 5-lipoxygenase (5-LOX), Phosphorylated extracellular signal regulated kinases (P-ERK), phosphorylated Jun N-terminal kinases (P-JNK), Mitogen-activated protein kinase (MAPK), P-P38, Toll-like receptor 4 (TLR4), Circulating endothelial cells (CEC), Alanine aminotransferase (ALT), Aspartate aminotransferase (AST), Total antioxidant capacity (T-AOC), Superoxide dismutase (SOD1), platelet-derived growth factor (PDGF), vascular cell adhesion molecule 1 (VCAM-1) were mentioned in one study.

## Quality assessment

Eleven studies [14, 15, 17–25] were graded as low in sequence generation as they declared that the animals were randomly divided into different groups. All studies had a low risk of baseline characteristics bias, as they described all animal characteristics and made sure that mice were similar in the baseline. The random and blinding outcome assessment of two studies [21, 25] was rated as low. All studies did not mention the allocation concealment, random housing, and blinded methods in a drug intervention. There were missing outcomes data in three studies [20–23]. The risk of selective outcomes reporting was high in three studies [15, 20, 23]. Across studies, the risk of bias from other sources was low. Overall, the methodological quality of the studies was relatively low. The quality of the studies and risk of bias assessments are presented in Figs 2 and 3.

**Table 2. Characteristics of all included studies.**

| Studies | Animals | Model(method) | Intervention and dose | Route | Duration | Anesthetic | Outcomes |
|---|---|---|---|---|---|---|---|
| Dong 2021 | SD rats (male, 6 weeks, 180 ±20g) 12/12/12/12 | high-fat diets 12 weeks + Vitamin D3 (2kg/ml, ip) | A: AS+TMP 50mg /kg<br>B: AS+TMP 100mg /kg<br>C: AS+TMP 200 mg /kg<br>D: AS+0.9% NS | i.p. | 4 weeks | 10% Chloral hydrate | 1. Body weight<br>2. TNF-α, IL-6, IL1β<br>3. TC, TG, LDL-C, AI<br>4. Vascular tension change value<br>5. VEGF, VEGFR-2<br>6. Histopathology (qualitative) |
| Zhao 2020 | Ldlr-/-hamsters (male, 8 weeks,/) 8/8 | high-cholesterol and high-fat diets 8 weeks | A: AS+TMPZ 32 mg/kg/d<br>B: AS+distilled water | i.g. | 8 weeks | 1% sodium phenobarbital (70 mg/kg) | 1. TG, TC, HDL<br>2. cAMP<br>3. Histopathology (quantify)(root)<br>4. platelet activity<br>5. PI3K, Akt, p-Akt |
| Yuan 2019 | ApoE-/-mice (male, 8 weeks, 20-25g) 6/6 | high-fat diets 12 weeks | A: AS+ TMP 5 mg/kg/d<br>B: AS+0.9% NS | i.p. | 4 weeks | / | 1. TC, TG, HDL-C, LDL-C, VLDL-C<br>2. VEGF, VEGFR2<br>3. Histopathology (quantify)(root)<br>4. CD31, vWF, HIF-1α, TNF-α |
| Zhang 2017 | ApoE-/-mice (male, 8 weeks, 18-20g) 10/10 | high-fat diets 12 weeks | A: AS+TMP 45.05 mg/kg<br>B: AS+distilled water | i.g. | 6 weeks | 0.1% sodium pentobarbital | 1. Histopathology (quantify)(root)<br>2. TG, TC, LDL-C<br>3. Bodyweight<br>4. Insulin Level<br>5. PAQR3, SCAP, SREBP1<br>6. IRS-1, PI3K, p-Akt, mTORC1 |
| Duan 2017 | ApoE-/-mice (male, 8 weeks, 22-24g) 10/10 | high-fat diets 16 weeks | A: AS+TMP 150mg/kg/d<br>B: AS+vehicle (20 ml/kg/d, 0.5% sodium carboxyl methyl cellulose) | i.g. | 8 weeks | CO2 | 1. ABCG1, ABCA1<br>2. SR-A, CD36<br>3. Histopathology (quantify)(root) |
| Ma 2015 | ApoE-/-mice (male, 8 weeks, 25.3 ±2g) 10/10 | high-fat diets 8 weeks | A:AS+TMP 100 mg/kg<br>B:AS+NS 10µl/g | i.p. | 8 weeks | 2% sodium pentobarbital | 1. Histopathology (quantify)(whole)<br>2. Aortic sinus pathology<br>3. SOD, GST,<br>4. Total and Nuclear Nrf-2 |
| Wang 2013 | New Zealand white rabbits (male, /, 1.0 ±0.15kg) 4/4/4 | high-cholesterol diet 12 weeks | A: AS+TMP 150mg/kg/d<br>B: AS+TMP 75 mg/kg/d<br>C: AS | i.g. | 12 weeks | sodium pentobarbital | 1. Histopathology (quantify)(whole)<br>2. the ratio of intimal/medial thickness<br>3. the number of monocytes in intimal<br>4. TC, TG, LDL-C, HDL-C<br>5. MCP-1, ICAM-1, LOX-1 |
| Dai 2013 | SD rats (male, 3-4weeks, 235±10g) 8/8/8 | high-fat diets 8 weeks +Vitamin D3 70U/kg*4 (i.p.) | A: AS+TMP 40mg/kg/d<br>B: AS+TMP 20mg/kg/d<br>C: AS | i.p. | 8 weeks | Urethane | 1. Histopathology (quantify)(whole)<br>2. Aortic sinus pathology<br>3.TLR4 |

(*Continued*)

**Table 2.** (Continued)

| Studies | Animals | Model(method) | Intervention and dose | Route | Duration | Anesthetic | Outcomes |
|---------|---------|---------------|----------------------|-------|----------|-----------|----------|
| Jiang 2011 | SD rats (male, /, 200g) 10/10/10 | atherogenic diets 6 weeks +vitamin D3 600,000IU/kg (i.p.) | A: AS+TMP 20mg/kg B: AS+TMP 80mg/kg C: AS+0.9% NS | i.g. | 6 weeks | 50mg/kg sodium pentobarbital | 1. TC, TG, LDL, HDL 2. SOD1, T-AOC, MDA 3. CEC, ALT, AST 3. body weight 4. Histopathology (quantify)(whole) |
| Lee 2009 | Wister rats (male, /, 200±20g) 8/8 | high-fat diets 40 days | A: AS+TMP 200mg/kg/d B: AS+distilled water | i.g. | 40 days | / | 1. PDGF 2. Histopathology (qualitative) |
| Zhang 2005 | Wister rats (male, /, 180-220g) 8/8 | high-fat diets 12 weeks | A: AS+TMP 200mg/kg/d B:AS | i.g. | 20 days | / | 1. TC, TG, LDL-C, HDL-C 2. Histopathology (qualitative) 3. VCAM-1 |
| Dong 1997 | Rabbits (male/female, 6-7months, 2.5 ±0.5kg) 12/12 | high-fat diets 10 weeks | A: AS+Ligustrazine Liposome 10ml (containing TMP 2mg/kg) B: AS | i.g. | 12 weeks | / | 1. TC, TG 2. MDA, SOD 3. Histopathology (qualitative) |

i.p. intraperitoneal injection; i.g. intragastric injection.

## Effectiveness

**Aortic atherosclerotic lesion area.** Meta-analysis of 10 comparisons from seven studies indicated that the atherosclerotic lesion area was to significantly decrease in TMP groups than in control groups (SMD = -2.16, 95% CI -3.08 to -1.25, $P$ = 0.0007, heterogeneity: $\chi 2$ = 28.77,

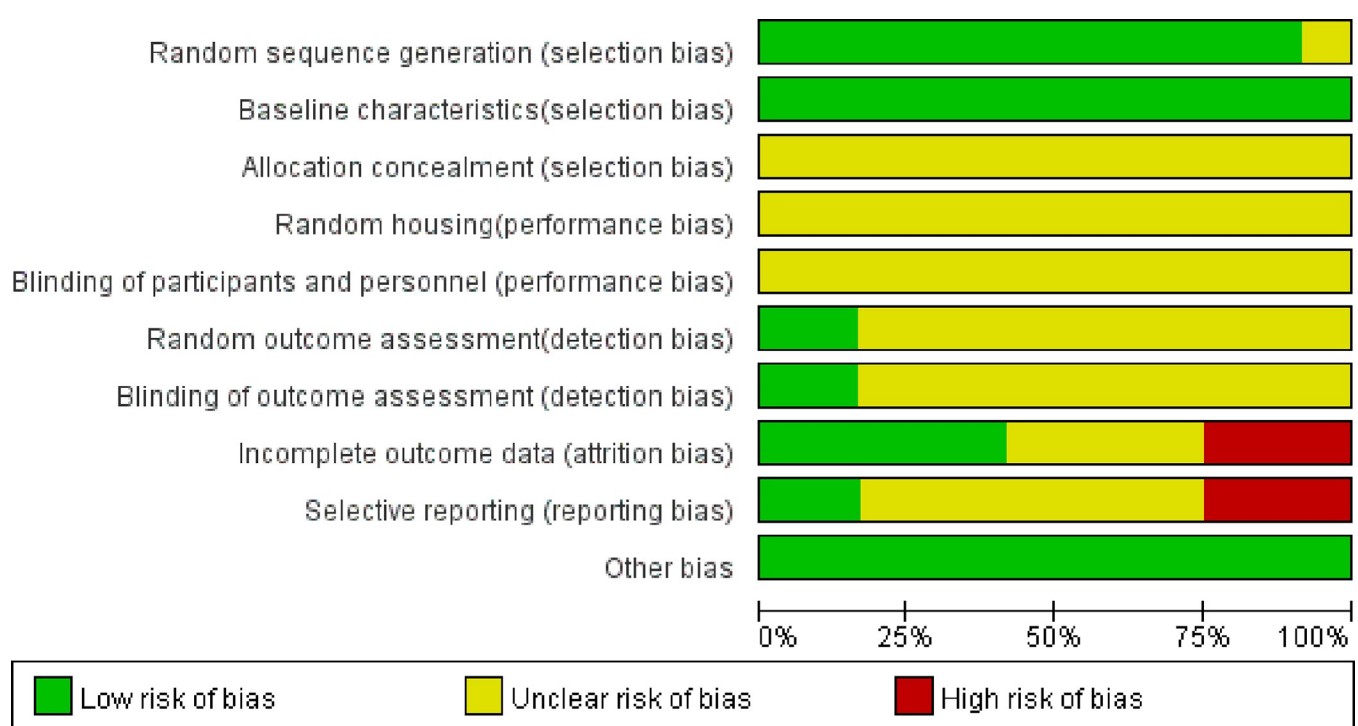

**Fig 2. Risk of bias and quality assessment.**

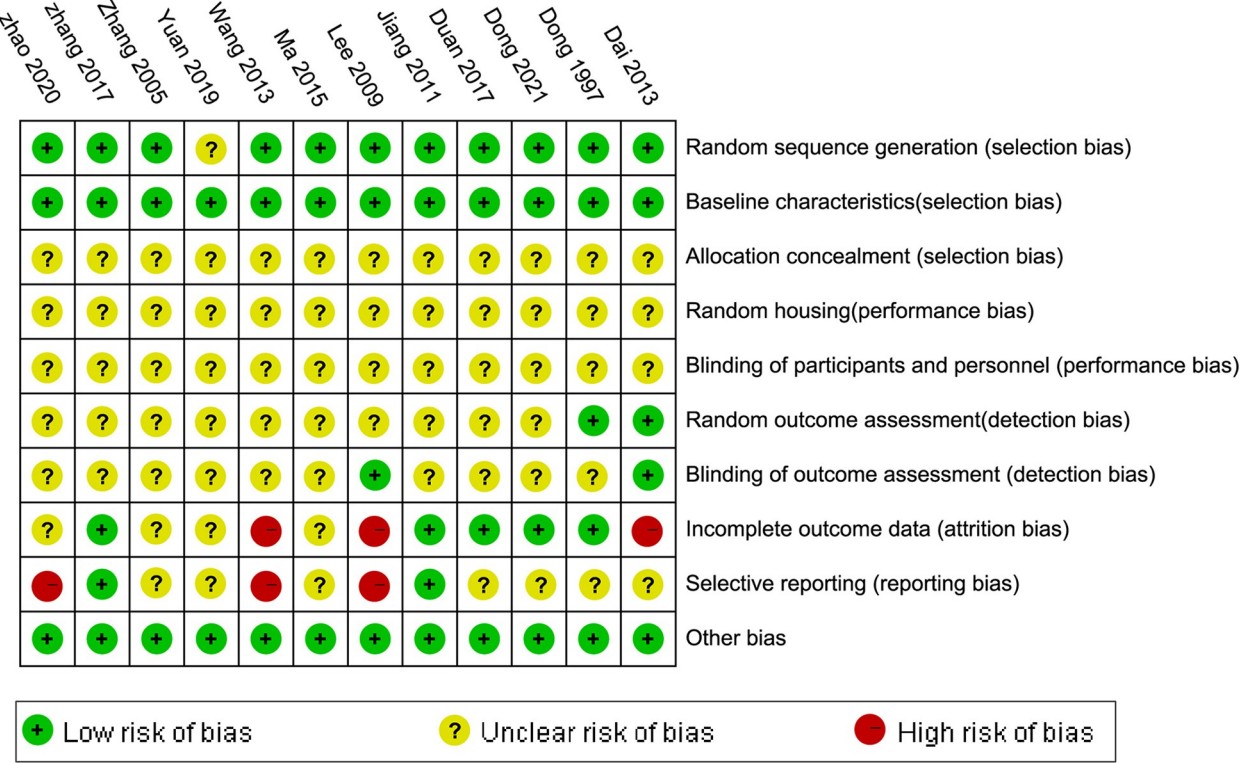

**Fig 3. The methodological quality of the included studies was assessed.**

$I^2 = 69\%$, Fig 4). In addition, the results of the other four qualitative studies included are as follows. Compared with the control group, TMP reduced the plaque deposits in the thoracic aorta in one study [14]. TMP reduced the average size of the atherosclerotic plaques in the aortic root in one study [18] and inhibited the thickening of intima smooth muscle and foam cell proliferation in two studies [23, 25].

**Plasma lipids.** Meta-analysis of 12 comparisons from eight studies indicated that TMP could significantly lower the concentration of TC (n = 184, SMD = -2.67, 95% CI -3.68 to -1.67, $P < 0.00001$, heterogeneity: $\chi2 = 55.29$, $I^2 = 80\%$, Fig 5) and TG (n = 184, SMD = -2.43,

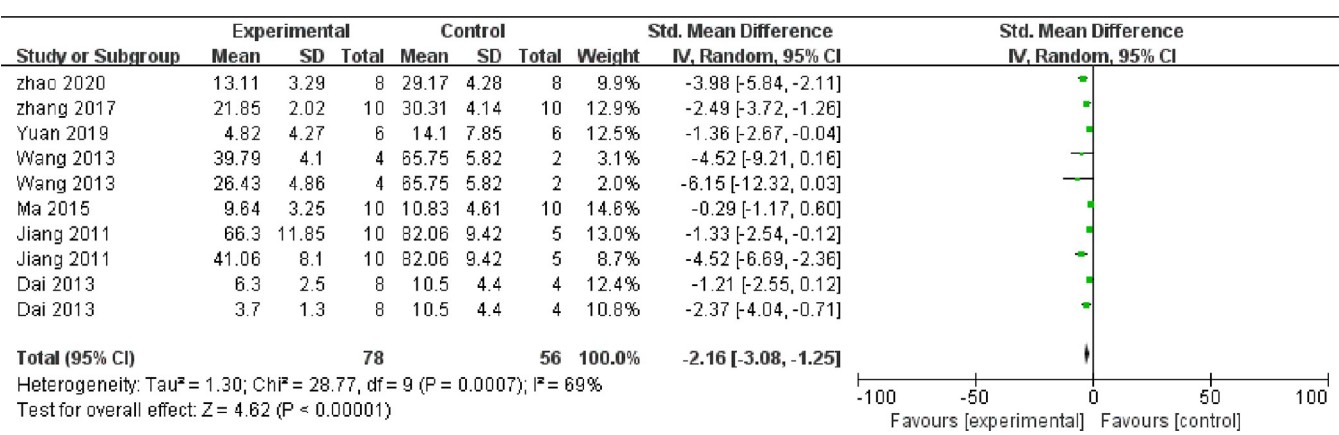

**Fig 4. Forest plot of TMP on aortic lesion area.**

**Fig 5. Forest plot of TMP on TC.**

95% CI -3.39 to -1.47, $P < 0.00001$, heterogeneity: $\chi2 = 54.03$, $I^2 = 80\%$, Fig 6). Meta-analysis of 8 comparisons from five studies indicated that LDL-C levels were lower in TMP groups than in control groups (n = 114, SMD = -2.87, 95% CI -4.16 to -1.58, $P < 0.00001$, heterogeneity: $\chi2 = 32.34$, $I^2 = 78\%$, Fig 7), while HDL-C levels were higher (n = 114, SMD = 2.04, 95% CI 1.05 to 3.03, $P = 0.001$, heterogeneity: $\chi2 = 24.24$, $I^2 = 71\%$, Fig 8). Two qualitative studies [15, 22] showed that TMP significantly increased in HDL levels.

**Inflammatory responses.** One study [14] showed that TMP significantly decreased plasma TNF-α, IL-1β, and IL-6 levels for plasma inflammatory responses, while the effect on TNF-α was useless in one study [16]. One study [21] indicated a significantly lower TLR-4 level.

**Anti-atherosclerotic mechanisms.** TMP-dose-dependent increases in VEGF and VEGFR-2 levels were seen in one study [14]. One study [15] found that TMP could significantly increase cAMP levels and decrease PI3K and P-Akt levels, leading to decreased platelet activity. In one study [16], TMP reduced VEGFR-2 levels, inhibited angiogenesis, and reduced CD31 and vWF expression. One study [17] showed that TMP down-regulated PAQR3,

**Fig 6. Forest plot of TMP on TG.**

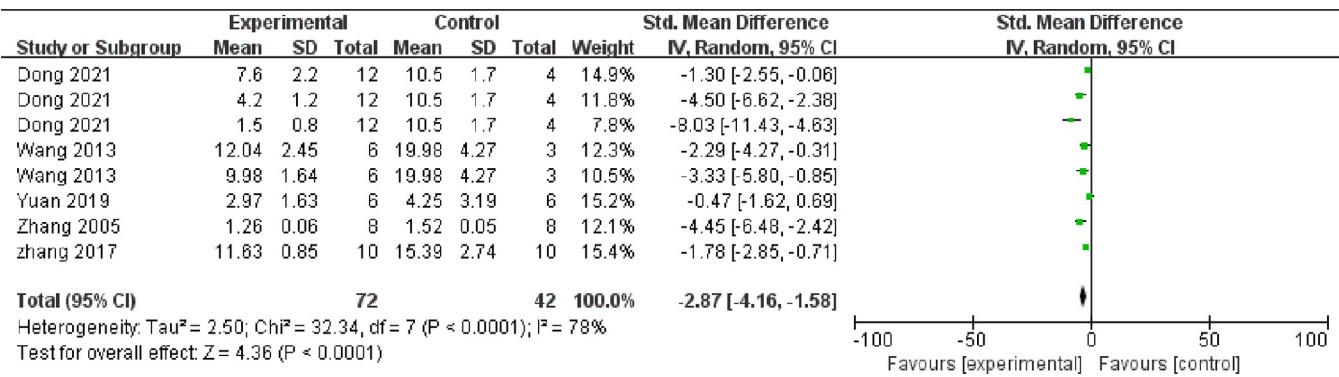

**Fig 7. Forest plot of TMP on LDL-C.**

subsequently down-regulated PI3K/Akt/mTORC1 signaling pathway, and inhibited the SCAP/ SREBP-1C signaling pathway reducing insulin and IRS-1 levels. One study [18] showed that TMP inhibited SR-A and CD36 expression and increased ABCA1 and ABCG1 expression through the inactivation of PI3K/Akt and P38 signaling pathways. Two studies [19, 25] indicated that TMP increased NRF-2 levels, SOD activity, GST levels, and reduced MDA levels to serve as antioxidants. According to one study [20], TMP inhibited the expression of LOX-1 and 5-LOX and P-ERK, P38, and PJNK MAPK induced by ox-LDL. One study reported [21] that TMP inhibited TLR-4 expression on arterial walls. TMP decreased blood CEC, ALT, and AST levels but restored blood T-AOC and SOD1 activities in one study [22]. According to one study [23], PDGF levels were reduced, which inhibited smooth muscle cell proliferation and migration.

## Subgroup analysis

The potential confounding factors (including various doses of TMP, different treatment durations, various assessing locations, and various types of animals) that may increase the heterogeneity of outcome measures were explored using stratified analysis of atherosclerotic lesion area. Subgroup analyses were conducted to investigate the effects of various doses on atherosclerotic lesion areas showing that the effect size increased with the dose of the drug. (SMD −1.3 vs. SMD −2.79 vs. SMD −3.33, $P$ = 0.03, Fig 9A). As expected, heterogeneity was remarkably reduced in low and medium dosages subgroups. At the same time, it remained high in the

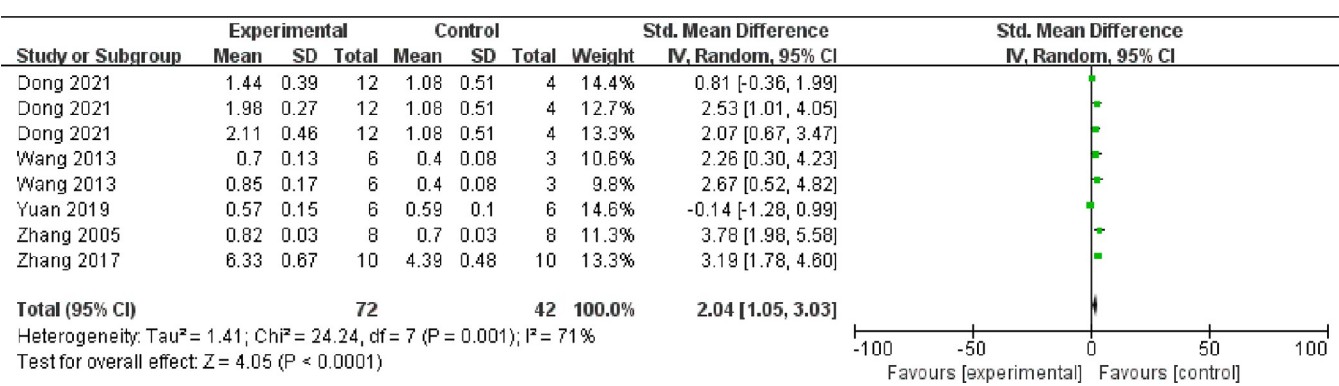

**Fig 8. Forest plot of TMP on HDL-C.**

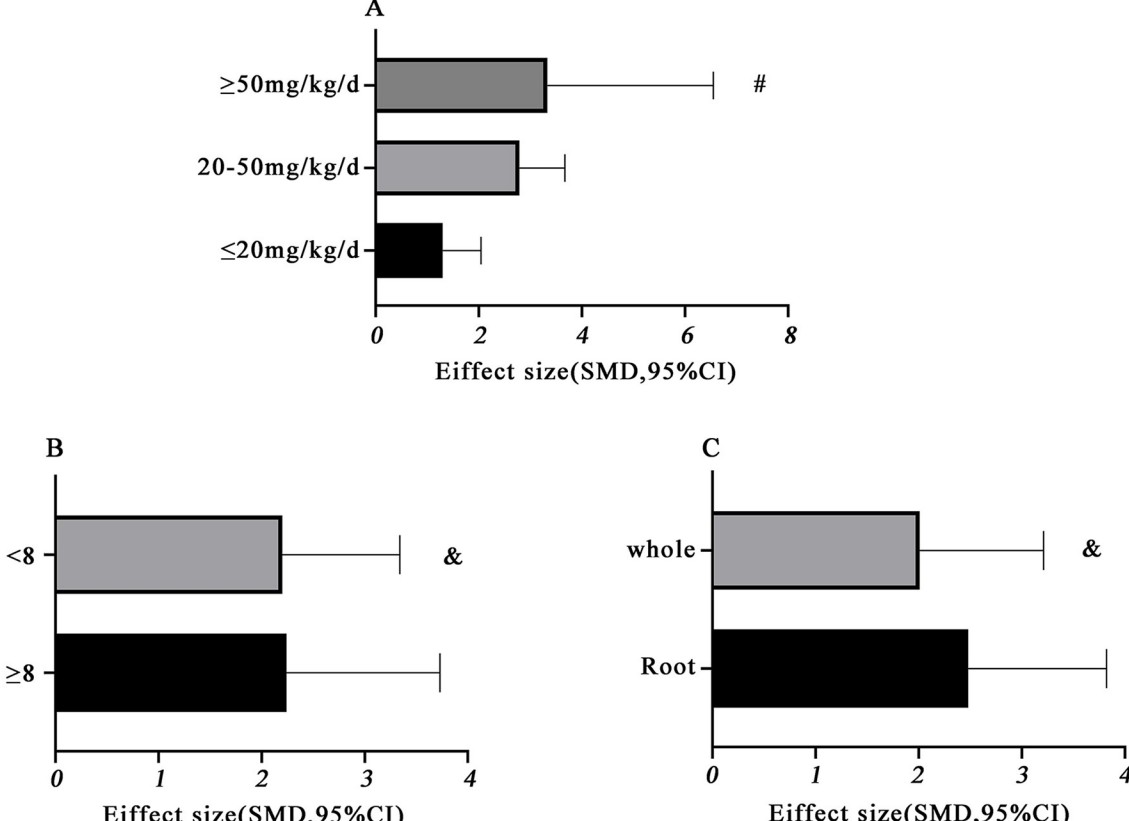

**Fig 9. Subgroup analyses of the aortic lesion area.** (A) The different doses of TMP on the effect size of the outcome measure; (B) the duration of treatment on the effect size of the outcome measure.; (C) the different locations of the aortic lesion of TMP on the effect size of the outcome measure. #$P < 0.05$ vs. control groups; &$P > 0.05$ vs. control groups.

subgroup of high dosage, which indicated that TMP dosage was the primary source of heterogeneity. No difference between the longer duration ($\geq 8$ w) of TMP treatment and the shorter treatment (<8w) (SMD −2.24 vs. SMD −2.20, $P = 0.97$, Fig 9B). There was no significant decrease in heterogeneity between the two groups. In the subgroup analysis of the various assessing location of the aortic lesion, no difference between the aorta root group and the whole aorta group(SMD −2.48 vs. SMD −2.01, $P = 0.61$, Fig 9C). There was no significant decrease in heterogeneity between the two groups. In addition to rats and mice, the other animals did not contain more than three studies. Thus subgroup analyses of various types of the animal were unable to be performed. We had not used Funnel plots to assess publication bias because fewer than ten studies involved a single outcome measure.

## Discussion

### Summary of main results

This is the first preclinical systematic review and meta-analysis to evaluate the therapeutic efficacy of TMP for animal models of AS. The result showed that TMP intervention could significantly decrease the aortic atherosclerotic lesion area, plasma lipids (TC, TG, and LDL-C), and some inflammatory levels and increase HDL-C levels in animal models. We also found a dose-response relationship between TMP and its therapeutic effect on the lesion area. The evidence suggests an anti-atherosclerosis role of TMP in animal models of AS via several mechanisms.

These findings could provide a scientific foundation for conducting clinical trials on TMP to treat cardiovascular diseases.

## Heterogeneity

Experimental designs and assessment methods varied greatly across preclinical studies, resulting in heterogeneity [26]. Our systematic review and meta-analysis identified the potential evidence of heterogeneity among the included studies by using subgroup analysis. In the subgroup analysis of ligustrazine dose, the effect size of the high-dose group ($\geq$50 mg /kg/d) was greater than that of the medium-dose group (20–50 mg /kg/d), and similarly, the medium-dose group was greater than that of the low-dose group ($\leq$20 mg) /kg/d), suggesting that different doses of TMP may be the source of heterogeneity. There was a positive correlation between drug dose and the reduction of the plaque area. According to the drug dose conversion formula between different administration methods and varieties, the optimal oral dose selected for the included studies was $\geq$50mg/kg/d. Considering that there is currently no comprehensive pathology-based study to investigate the dose-response relationship of TMP to AS, there is an urgent need to determine the optimal dose and the safety and toxicity of TMP. One of the potential sources of heterogeneity was believed to come from assessing the location of the aortic lesion differently. These plaques preferred to appear in the aortic root. In humans, atherosclerotic plaques tend to occur in coronary arteries, whereas in mice, atherosclerotic plaques cluster in the aortic root [27]. However, in the subgroup analysis of the effects of different measurement sites and different courses of treatment on the atherosclerotic plaque, we found no significant difference in effect size. The species significantly impacted the aortic atherosclerotic lesion area and plasma lipids. ApoE-/- mice and Ldlr-/- mice are mainly used for AS. ApoE-/- is the most suitable model for studying AS. It is a spontaneous AS model, which is more similar to AS in humans, and its TC level does not change with gender and age. AS in Ldlr-/- mice is not present in a regular diet and requires high cholesterol diet induction. Atherosclerosis can be developed in both mice after 12 weeks [28]. This meta-analysis did not have sufficient statistical quantity to compare the effects on arterial plaque area of different animal species. More studies are needed to allow a meta-analysis of the effects on various animal species.

## Possible cellular and molecular mechanisms of TMP against AS

Atherosclerosis is associated with dyslipidemia, including elevated TG, TC, LDL-C levels, and a low HDL-C level [29]. Dynamic lipid imbalance is closely associated with AS pathogenesis, which has become a critical determinant of the occurrence and development of AS. Concerning plaque area, TC, and TG, the favorable efficacy results of this meta-analysis were consistent with the results of each study in this review and statistically confirmed the significant improvement in AS by TMP. One result was inconsistent with the overall results in the meta-analyses for LDL-C and HDL-C, leading to an increase in heterogeneity. Notably, not all lipid profiles would be affected in our findings, which indicated that there might be particular pathways through which TMP decreased the plasma lipid level. Most lipids are metabolized in the liver and adipose tissue [30]. The SREBP and its escort protein SCAP are involved in cholesterol biosynthesis [31]. It is crucial to the synthesis of AS [32]. PAQR3 belongs to the progestin and adipoQ receptors (PAQRs) family. Research suggested that PAQR3 interacts with SCAP and SREBP, promoting the formation of the SCAP/SREBP complex, increasing SREBP processing, and promoting lipid synthesis [31]. By regulating SREBP activation, the PI3K/Akt/mTORC1 pathway helps maintain lipid raft integrity [33]. It is thought that TMP can alleviate lipid metabolism disorders through downregulating PAQR3 and inhibiting SCAP/SREBP-1c

signaling pathways. PI3K/Akt/mTORC1 may play a role [17]. Insulin resistance is a critical factor in the development of AS. Additionally, TMP can ameliorate insulin resistance and inhibit the expression of Insig-1 in adipose tissue of ApoE-/- mice fed with a high-fat diet [17].

The atheroprotective effect of HDL is attributed to reverse cholesterol transport (RCT), which promotes the efflux of excess cholesterol from macrophage-derived foam cells [34]. TMP may reduce ox-LDL uptake and increase cholesterol expulsion in macrophage-derived foam cells. The primary receptors for OX-LDL uptake are CD36 and SR-A, while the principal receptors for cholesterol efflux are ABCG1 and ABCA1 [35]. Inactivating PI3K signaling by TMP decreased CD36 and SR-A expression and OX-LDL uptake [18]. In addition, TMP increased ABCA1 expression and ABCA1-mediated cholesterol efflux via inhibition of P38 signaling [18].

Existing network pharmacological studies have confirmed that TMP mainly inhibits inflammatory response through multi-target coordination and regulation of multiple signaling pathways, thus treating AS [36]. TLR4 signaling pathways are essential in activating and amplifying inflammatory responses in AS. Activated TLR4 signaling pathways can stimulate the secretion of inflammatory molecules such as TNF-α, IL-6, IL-8, IL-12, IL-23, and IL-1β [37]. This review suggests that TMP inhibits inflammation in atherosclerosis by modulating TLR4, IL-6, and IL-1β levels. Atherosclerosis begins with the recruitment of leukocytes onto the endothelium wall of the vessel [38]. During the recruitment of white blood cells from the bloodstream to the intima of the vessel, Icam-1 primarily acts as an integrin receptor, whereas MCP-1 helps recruit monocytes [35]. The pro-inflammatory cytokines induce the expression of ICAM-1, MCP-1, LOX-1, and 5-LOX and convert macrophages and smooth muscle cells into foam cells to form atherosclerotic plaques [39]. Additionally, these pro-inflammatory processes mediated by OX-LDL and LOX1 are thought to result in enlargement of lipid cores, rupture of lesions, and instability of arterial thrombosis [40]. TMP can ameliorate this process by down-regulating the expression of LOX-1 and 5-LOX and inhibiting the expression of ICAM-1 and MCP-1 [20].

In the initiation and progression of atherosclerosis, oxidative stress plays a critical role [41]. Nrf-2 is a transcription factor related to antioxidant stress that increases endogenous antioxidant activity, reducing oxidative stress in the body [42]. The most common antioxidant indicators are SOD, MDA, GSH-PX, and T-AOC [43]. In this review, we found that TMP can promote the expression of NRF-2 [19], further promote the expression of downstream antioxidant indices such as SOD, GST, and T-AOC, inhibit the expression of MDA [25], and reduce the risk of atherosclerosis caused by oxidative stress.

Vascular injury and endothelial apoptosis play crucial roles in the pathogenesis of atherosclerosis [44]. As an essential pro-angiogenic factor, VEGF can act on endothelial cells, promote endothelial cell mitosis, increase vascular permeability, and play a role in the early stage of the angiogenesis cascade reaction. Reduced VEGF expression can lead to vascular degradation, bleeding, and plaque rupture [45]. The levels of VEGF and VEGFR-2 in thoracic aorta vessels of rats in ligustrazine dose groups increased significantly in a dose-dependent manner [14], suggesting that ligustrazine can improve the integrity of rat vascular endothelial, thereby improving the vascular endothelial function of rats, and exerting its vascular protection effect. Activation of the ERK signaling pathway is critical for cardiovascular protection [46]. P38, as well as ERK and JNK, constitute the MAPK signaling pathway [47]. P38 signaling pathway inhibits ERK signaling pathway in apoptosis and free radical injury models in vitro. Additionally, TMP inhibited OX-LDL-induced activation of p-ERK, p38, and p-JNK MAPK [20].

Hyperactivity of the platelets and an increase in coagulation are significant factors that contribute to the development of atherosclerosis and thrombosis [48]. CAM can mediate the recognition and binding between cells and extracellular matrix and some plasma proteins, and

the adhesion and aggregation of platelets. VCAM-1 is a kind of CAM. As one of its mechanisms to combat AS, TMP inhibited platelet aggregation and downregulates the expression of VCAM-1 [24].

The proliferation and migration of smooth muscle cells to intima is the primary pathological process of AS [49]. Many growth factors mediate the migration of smooth muscle cells to the intima, and PDGF is one of these growth factors [50]. PDGF promotes AS by causing vascular smooth muscle cells to migrate and proliferate [51]. PDGF appears in different stages of human AS, and it is crucial to the development and formation of AS. Hence, inhibiting the expression of PDGF may effectively block AS. TMP can reduce serum PDGF levels [23] and inhibit smooth muscle cell migration and proliferation in AS rats.

AS plaque instability can be caused by angiogenesis, and inhibition of angiogenesis can stabilize AS plaque. VEGFR-2 plays a vital role in angiogenesis, and increased VEGFR-2 significantly promotes angiogenesis within plaques, leading to plaque instability [52]. Immunofluorescence staining was used to label specific markers such as CD31 and vWF to evaluate angiogenesis [53]. It was found that the density of neovascularization in plaque increased in AS model mice, and TMP could reduce the expression of CD31 but had no significant effect on the mature vascular marker vWF [16], suggesting that TMP may inhibit angiogenesis in plaque but has no significant effect on mature vessels.

By summarizing the included studies, The possible mechanisms of TMP on the therapeutic efficacy of AS are as follows:(1) ameliorate lipid metabolism disorder via downregulating PAQR3 and inhibiting the SCAP/SREBP-1c signaling pathway. In addition, PI3K/Akt/mTORC1 signaling pathway may be involved in this process [17] by suppressing lipid accumulation in macrophages via PI3K/Akt and p38 MAPK signaling to downregulate scavenger receptors and upregulate ATP-binding cassette transporters [18]. (2) antioxidant effects via increasing Nrf-2 to enhance the activity of SOD, GST [19, 25] while decreasing the MDA generation and inhibiting the induction of antioxidant genes both in the aorta and in the liver [22]. (3) anti-inflammation via decreasing the expression of TLR4 and preventing macrophages from forming foam cells [20, 21]. (4) protection of endothelial function via regulating the VEGF signaling pathway [14] and suppressing ox-LDL-induced activations of p-ERK, p-p38, and p-JNK MAPK [20]. (5) antiplatelet activity via down-regulating VCAM-1 [24] and inhibiting PI3K/Akt, cAMP, and calcium signal pathways [15]. (6) reduces smooth muscle cells proliferation and migration via down-regulating PDGF [23]. (7) inhibits angiogenesis within plaques via down-regulation of VEGFR2 [16]. The mechanism diagram is shown in Fig 10.

## Limitation

Published studies lacking atherosclerotic plaque area data, which were not included in our coverage, may also lead to selection bias. A heterogeneity test was carried out on the results of the included experimental studies. We found that the results of most outcome indicators had high heterogeneity, which may be due to differences in the modeling time of animal species and gender and the detection time of dietary indicators among the studies. However, subgroup analysis was not conducted due to the limited number of included studies. The quality of the included studies we analyzed was poor. The sample sizes of some included studies were relatively small. It could also be a source of heterogeneity, affecting our conclusions. As many items in the risk and quality assessment component are unclear, further efforts are needed to standardize the design and implementation of animal intervention experiments. Some indicators, such as plasma inflammatory response and other anti-atherosclerosis indicators of ligustrazine, were not reported in more than three studies, so meta-analysis was impossible. In

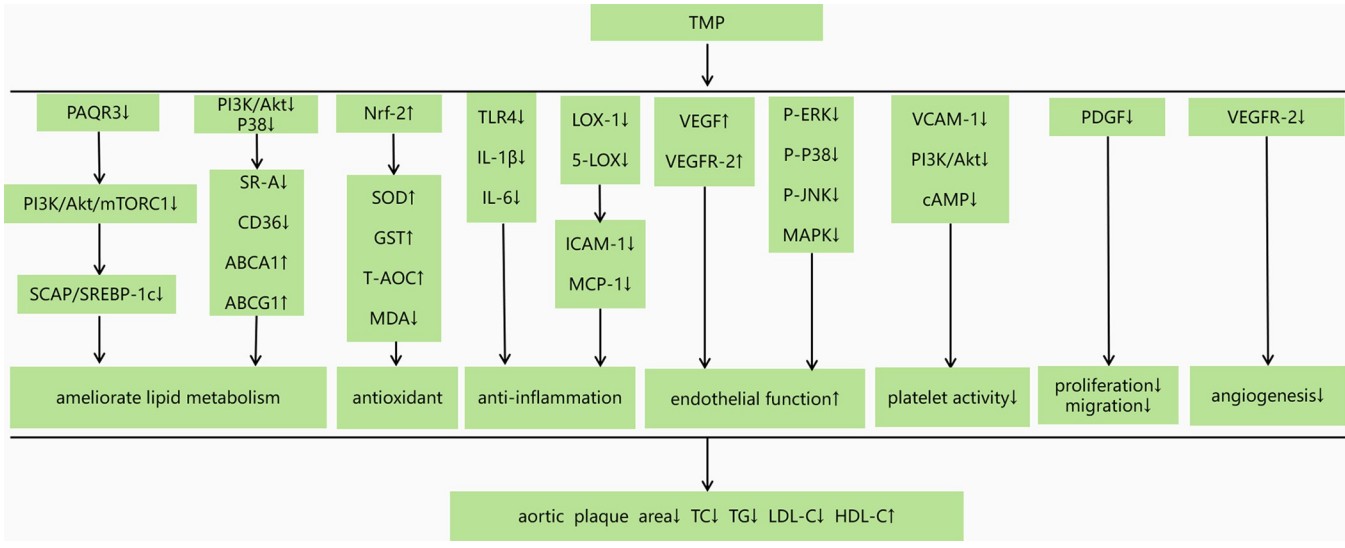

**Fig 10. A schematic representation of mechanisms of TMP for AS.**

addition, TMP intervention in animals begins before the onset of disease, while the onset of treatment in humans is difficult to achieve.

## Conclusion

The findings of the present study suggest that TMP exerts anti-atherosclerosis functions in an animal model of AS mediated by anti-inflammatory action, antioxidant action, ameliorate lipid metabolism disorder, protection of endothelial function, antiplatelet activity, reducing the proliferation and migration of smooth muscle cells, inhibition of angiogenesis, antiplatelet aggregation. It should be noted that the antiatherogenic effect significantly correlated with the dose of TMP, with higher doses of TMP($\geq$50mg/kg/d) proving to be more effective. Because of methodological flaws, positive conclusions should be treated with caution. Standardized designing guidelines in preclinical studies of AS are recommended. Nevertheless, our findings suggest that TMP is a candidate drug for the treatment of AS.

## Supporting information

**S1 Checklist.**
(DOC)

## Author Contributions

**Conceptualization:** SiJin Li, Ping Liu, XiaoTeng Feng, YiRu Wang.

**Data curation:** SiJin Li, Ping Liu, XiaoTeng Feng, YiRu Wang, Min Du, JiaRou Wang.

**Formal analysis:** SiJin Li, Ping Liu, XiaoTeng Feng, Min Du.

**Investigation:** SiJin Li.

**Methodology:** SiJin Li, Ping Liu, XiaoTeng Feng, YiRu Wang, Min Du, JiaRou Wang.

**Resources:** SiJin Li, YiRu Wang.

**Software:** SiJin Li, XiaoTeng Feng.

**Supervision:** Ping Liu.

**Visualization:** SiJin Li.

**Writing – original draft:** SiJin Li.

**Writing – review & editing:** SiJin Li, Ping Liu, XiaoTeng Feng, YiRu Wang, Min Du, JiaRou Wang.

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
