## [Decision Letter · Decision Letter 0]

23 Mar 2022

PONE-D-22-00534The role and mechanism of tetramethylpyrazine for atherosclerosis in animal models: A systematic review and meta-analysis.PLOS ONE

Dear Dr. Liu,

Thank you for submitting your manuscript to PLOS ONE. After careful consideration, we feel that it has merit but does not fully meet PLOS ONE’s publication criteria as it currently stands. Therefore, we invite you to submit a revised version of the manuscript that addresses the points raised during the review process.

We look forward to receiving your revised manuscript.

Kind regards,

Michael Bader

Academic Editor

PLOS ONE

Journal Requirements:

6. Please include a caption for figure 10.

Reviewers' comments:

Reviewer's Responses to Questions

**Comments to the Author**

1. Is the manuscript technically sound, and do the data support the conclusions?

Reviewer #1: Yes

Reviewer #2: Yes

2. Has the statistical analysis been performed appropriately and rigorously? 

Reviewer #1: Yes

Reviewer #2: Yes

3. Have the authors made all data underlying the findings in their manuscript fully available?

Reviewer #1: Yes

Reviewer #2: Yes

4. Is the manuscript presented in an intelligible fashion and written in standard English?

Reviewer #1: No

Reviewer #2: Yes

5. Review Comments to the Author

Reviewer #1: This study gave a systematic review and meta-analysis of tetramethylpyrazine for the treatment of atherosclerosis in animal models. Although, limitations of the quantity and quality of previous studies exist, tetramethylpyrazine displayed therapeutic beneficial effects in atherosclerosis treatment. This study will provide useful information for those who works in the same areas. However, this manuscript was badly written and needs minor revision before publication.

1. On page 5 line 99, the word "atherosclerosis" is repeated and should be deleted.

2. On page 6, "type of participants", it is advised that AS models with diabetes, hypertension, and other diseases were included and analyzed since AS often associated with these diseases.

3. In many places in the manuscript, the space between punctuation and the next sentence is omitted.

Reviewer #2: The manuscripts titled”The role and mechanism of tetramethylpyrazine for atherosclerosis in animal models: A systematic review and meta-analysis ” presents that TMP exerts anti-atherosclerosis functions in animal models of AS by a systematic review and meta-analysis. The research were well designed to support the conclusion. However, some questions needed to be concerned by authors. For the authors’ convenience, I list my suggested revision as following:

1.Please correct the spaces mistakes throughout the manuscript.

2.Please maker sure that such descriptions like “ApoE-/-” and “Ldlr-/-”were always consistent in the manuscript.

3.In “Introduction” section, please rewrite the descriptions such as “This study systematically evaluated and meta-analyzed the effects of ligustrazine on animal models of atherosclerosis and further discussed the underlying cellular and molecular mechanisms so that it can better guide clinical practice and clinical trial design” to shorter sentences.

4.In results section, please describe each figures and write the results only. Please rewrite it and modify it. Delete irrelevant sentences and use more scientific language.

5.The format of Table1 was different with Table2, please check it and modify it.

6.On page 12 line 195, there is no need to enter a new line.

7.In section of “Anti-atherosclerotic mechanisms ” on Line 266, add more relevance sentences about correlation within the factors.

8.The discussion parts needs illustrated the object of the study. However, authors show more information about AS which have been clearly elaborated in your introduction parts. Therefore, the authors should expand the description about your present work by citing updated references. The discussion needs to replace with a better one which was well within the Objective of your manuscript.

9.In Fig.10, there needs to correct the mistake of “P-EPK” to “P-ERK” and make sure that the abbreviations are consistent with your manuscript.

10.The manuscript needs to write clearly about your results and lightspot, and less repetitious point.

6. PLOS authors have the option to publish the peer review history of their article (what does this mean?). If published, this will include your full peer review and any attached files.

Reviewer #1: No

Reviewer #2: No

---

## [Author Response · Author response to Decision Letter 0]

10 Apr 2022

RESPONSE TO REVIEWER #1: 

This study gave a systematic review and meta-analysis of tetramethylpyrazine for the treatment of atherosclerosis in animal models. Although, limitations of the quantity and quality of previous studies exist, tetramethylpyrazine displayed therapeutic beneficial effects in atherosclerosis treatment. This study will provide useful information for those who works in the same areas. However, this manuscript was badly written and needs minor revision before publication.

1. On page 5 line 99, the word "atherosclerosis" is repeated and should be deleted.

Author response: Thank you for that observation. We have corrected the word“atherosclerosis” for “atherogenesis” and checked that there are no more spelling mistakes. 

2. On page 6, "type of participants", it is advised that AS models with diabetes, hypertension, and other diseases were included and analyzed since AS often associated with these diseases.

Author response: We thank reviewer #1 for pointing out this problem, and we agree with reviewer #1 that diabetes, hypertension, and other diseases are related to atherosclerosis. We reviewed the literature of 33 non-atherosclerotic models excluded again and found that there were 1 model of endothelial hyperplasia after arterial deendothelium, 1 model of intimal hyperplasia after coronary artery balloon injury, 1 model of spontaneous hypertension in rats, 7 models of vascular dementia, and 2 models of myocardial ischemia. There were 1 myocardial infarction model, 1 rabbit iliac atherosclerotic stenosis model, 3 diabetes model, 2 diabetic nephropathy models, 2 ischemic stroke models, 3 cerebral thromboembolism models, 3 chronic renal failure atherosclerosis model, 2 coronary artery stenosis model, and 4 hyperlipidemic models. Some of these models are accompanied by atherosclerosis, but their studies lack major detection indicators of atherosclerosis (the histopathological analysis of the atherosclerotic lesion area) and mainly focus on other diseases, which do not meet our inclusion criteria for detection indicators. so we cannot include them in this study. Perhaps we can use these diseases as models for systematic review and meta-analysis in future studies.

3. In many places in the manuscript, the space between punctuation and the next sentence is omitted.

Author response: We thank reviewer #1 for the patience and suggestion. The manuscript has been checked thoroughly again and some spaces errors were corrected.

RESPONSE TO REVIEWER #2:

The manuscripts titled “The role and mechanism of tetramethylpyrazine for atherosclerosis in animal models: A systematic review and meta-analysis ” presents that TMP exerts anti-atherosclerosis functions in animal models of AS by a systematic review and meta-analysis. The research were well designed to support the conclusion. However, some questions needed to be concerned by authors. For the authors’ convenience, I list my suggested revision as following:

1.Please correct the spaces mistakes throughout the manuscript.

Author response: We thank reviewer #2 for the patience and suggestion. We have carefully revised the manuscript, and the spaces mistakes were corrected.

2. Please maker sure that such descriptions like “ApoE-/-” and “Ldlr-/-”were always consistent in the manuscript.

Author response: Thank you for your careful observation. This has been corrected.

3. In “Introduction” section, please rewrite the descriptions such as “This study systematically evaluated and meta-analyzed the effects of ligustrazine on animal models of atherosclerosis and further discussed the underlying cellular and molecular mechanisms so that it can better guide clinical practice and clinical trial design” to shorter sentences.

Author response: We have rephrased this sentence (lines 85-87). Thank you.

4. In results section, please describe each figures and write the results only. Please rewrite it and modify it. Delete irrelevant sentences and use more scientific language.

Author response: We are grateful for this suggestion. We have added the description of the figures and deleted irrelevant sentences. We have tried our best to polish the language in the revised manuscript and hoped that the results are clearly presented.

5. The format of Table1 was different with Table2, please check it and modify it.

Author response: Thank you for pointing this out. We corrected and unified the format of the tables.

6. On page 12 line 195, there is no need to enter a new line.

Author response: Changed as requested. Thank you.

7. In section of “Anti-atherosclerotic mechanisms ” on Line 266, add more relevance sentences about correlation within the factors.

Author response: Thank you for your useful and constructive comments. We have rewritten the section on “Anti-atherosclerotic mechanisms” (on Line 274-291) to make it easier to understand.

8. The discussion parts needs illustrated the object of the study. However, authors show more information about AS which have been clearly elaborated in your introduction parts. Therefore, the authors should expand the description about your present work by citing updated references. The discussion needs to replace with a better one which was well within the Objective of your manuscript.

Author response: The reviewer makes an excellent point. In re-reading the paper, it’s clear that we were overly focused on presenting the data and did an embarrassingly poor job of referencing much of the recent literature. We again regret this oversight and have extensively rewritten the discussion and amended the manuscript to include a much more comprehensive discussion of recently published work. 

9. In Fig.10, there needs to correct the mistake of “P-EPK” to “P-ERK” and make sure that the abbreviations are consistent with your manuscript.

Author response: We thank reviewer #2 for pointing this error out. We made this correction and others in the revised manuscript.

10. The manuscript needs to write clearly about your results and lightspot, and less repetitious point.

Author response: We apologize for not more clearly describing these results. We have seriously revised the manuscript to emphasize the highlights and ensure that there is less repetition. we hoped that the current version could be better understood.

---

## [Decision Letter · Decision Letter 1]

20 Apr 2022

The role and mechanism of tetramethylpyrazine for atherosclerosis in animal models: A systematic review and meta-analysis.

PONE-D-22-00534R1

Dear Dr. Liu,

We’re pleased to inform you that your manuscript has been judged scientifically suitable for publication and will be formally accepted for publication once it meets all outstanding technical requirements.

Kind regards,

Michael Bader

Academic Editor

PLOS ONE

Additional Editor Comments (optional):

Reviewers' comments:

Reviewer's Responses to Questions

**Comments to the Author**

1. If the authors have adequately addressed your comments raised in a previous round of review and you feel that this manuscript is now acceptable for publication, you may indicate that here to bypass the “Comments to the Author” section, enter your conflict of interest statement in the “Confidential to Editor” section, and submit your "Accept" recommendation.

Reviewer #2: All comments have been addressed

2. Is the manuscript technically sound, and do the data support the conclusions?

Reviewer #2: Yes

3. Has the statistical analysis been performed appropriately and rigorously? 

Reviewer #2: Yes

4. Have the authors made all data underlying the findings in their manuscript fully available?

Reviewer #2: Yes

5. Is the manuscript presented in an intelligible fashion and written in standard English?

Reviewer #2: Yes

6. Review Comments to the Author

Reviewer #2: (No Response)

7. PLOS authors have the option to publish the peer review history of their article (what does this mean?). If published, this will include your full peer review and any attached files.

Reviewer #2: No

---

## [Editor Report · Acceptance letter]

22 Apr 2022

PONE-D-22-00534R1 

The role and mechanism of tetramethylpyrazine for atherosclerosis in animal models: A systematic review and meta-analysis 

Dear Dr. Liu:

I'm pleased to inform you that your manuscript has been deemed suitable for publication in PLOS ONE. Congratulations! Your manuscript is now with our production department. 

Kind regards, 

on behalf of

Prof. Michael Bader 

Academic Editor

PLOS ONE